# Artificial trans-kingdom RNAi of *FolRDR1* is a potential strategy to control tomato wilt disease

**Shou-Qiang Ouyang**[1,2]*, **Hui-Min Ji**[2], **Tao Feng**[2], **Shu-Jie Luo**[2], **Lu Cheng**[2], **Nan Wang**[2]

**1** College of Life Sciences, Zhejiang Normal University, Jinhua, China, **2** College of Horticulture and Plant Protection, Yangzhou University, Yangzhou, China

* sqouyang@zjnu.edu.cn

## Abstract

Tomato is cultivated worldwide as a nutrient-rich vegetable crop. Tomato wilt disease caused by *Fusarium oxysporum* f.sp. *Lycopersici* (*Fol*) is one of the most serious fungal diseases posing threats to tomato production. Recently, the development of Spray-Induced Gene Silencing (SIGS) directs a novel plant disease management by generating an efficient and environmental friendly biocontrol agent. Here, we characterized that *FolRDR1* (RNA-dependent RNA polymerase 1) mediated the pathogen invasion to the host plant tomato, and played as an essential regulator in pathogen development and pathogenicity. Our fluorescence tracing data further presented that effective uptakes of *FolRDR1*-dsRNAs were observed in both *Fol* and tomato tissues. Subsequently, exogenous application of *FolRDR1*-dsRNAs on pre-*Fol*-infected tomato leaves resulted in significant alleviation of tomato wilt disease symptoms. Particularly, *FolRDR1*-RNAi was highly specific without sequence off-target in related plants. Our results of pathogen gene-targeting RNAi have provided a new strategy for tomato wilt disease management by developing an environmentally-friendly biocontrol agent.

**Data Availability Statement:** The raw sequence data for this study are available in the National Genomics Data Center with accession no.

## Author summary

Tomato wild disease caused by *Fusarium oxysporum* f. sp. *lycopersici* (*Fol*) threatens the global tomato production for both processing and the fresh-market system. An efficient and eco-friendly strategy to control wilt disease is urgent needed. SIGS is a novel discovered RNA silencing strategy for disease control, and effectively used for crops protection based on the uptake of external dsRNA by plant pathogens. By homologous comparison. We identified *FolRDR1* in the existing gene annotation of *Fol*. Here, our data showed that exogenous application of *FolRDR1*-dsRNAs on pre-*Fol*-infected tomato leaves resulted in significant alleviation of tomato wilt disease symptoms. Particularly, *FolRDR1*-RNAi was highly specific without sequence off-target in related plants.

CRA011174. https://bigd.big.ac.cn/gsa/browse/CRA011174.

**Funding:** This work was partially supported by a grant from the National Natural Science Foundation of China #31972351 and a grant from Zhejiang Natural Science Foundation #KYZ34423025 to S.Q.O. The funders had no role in study design, data collection and analysis, decision to publish, or preparation of the manuscript.

**Competing interests:** The authors have declared that no competing interests exist.

## Introduction

Tomato (*Solanum lycopersicum* L.) is one of the most important vegetable worldwide. Because tomato is susceptible to more than 200 pests and microbe pathogens, disease control for appropriate disease resistances is crucial to the commercial production [1]. Fusarium wilt of tomato, caused by *Fusarium oxysporum* f. sp. *lycopersici* (*Fol*), threatens the global tomato production for both processing and the fresh-market system. *Fol* penetrates tomato roots before colonizing the vascular tissue. Initial disease symptoms are visible at the early stage about one week with wilting of lower basal leaves, acropetally to the upper leaves until the entire plant die. In the absence of control strategies, such as fumigation and host resistance, Fusarium wilt disease can causes up to complete tomato crop loss [2]. Therefore, an efficient and eco-friendly strategy to control wilt disease is urgent needed.

Crops are always challenged by environmental stresses from different kinds of parasites, such as bacteria, fungi, viruses, oomycetes, insects, and parasitic plants throughout their life cycles. To maintain surveillance of progenies, plant hosts have evolved fine-tuned defense mechanisms [3,4,5]. In eukaryotic world, RNA interference (RNAi), triggered by small RNAs (sRNAs) such as small interfering RNAs (siRNAs) and miRNAs, is an evolutionarily conserved and sequence-specific mechanism that regulates target gene expression at either the transcriptional level (transcriptional gene silencing, TGS) or the posttranscriptional level (posttranscriptional gene silencing, PTGS) [6,7,8,9,10]. So far, different RNAi pathways including Host-Induced Gene Silencing (HIGS), Virus-Induced Gene Silencing (VIGS) and Spray-Induced Gene Silencing (SIGS) are reported for the artificial silencing of genes [11]. SIGS is a novel discovered RNA silencing strategy for disease control, and effectively used for crops protection based on the uptake of external dsRNA by plant pathogens [12,13,14]. Therefore, SIGS is a valuable eco-friendly and advanced innovative strategy for plant disease control at pre-harvesting and post-harvesting stages but with limited off-target effects [15].

The RNAi silencing process is initiated with long dsRNAs being cleaved into small fragments of sRNAs (21–25 nts) by the Dicer (DCL), followed by loading into the RNA Induced Silencing Complex (RISC) which contains an essential member such as Argonaute (AGO) protein [16]. The mechanism is fulfilled by amplifying sRNA molecules through RNA-dependent RNA polymerase (RDRs) through the amplification of double-stranded RNA (dsRNA), which are further cleaved and processed by DCLs and trigger the next round of RNAi [17]. A crucial role of RDRs is to interact with RNAi machinery and provide defense against pathogens. RDRs are evolutionarily distributed into four subclasses (RDR1, 2, 3, and 6) in plants [18]. Similarly, RDRs are also widely distributed in three major groups of fungi *Ascomycetes*, *Basidiomycetes* and *Zygomycetes* [18]. However, no *RDR*-like gene has been reported in *F. oxysporum* so far. In this study, we tried to find the genes, such as *DCL*, *RDR*, *AGO* and other gene families, related to the production and function of sRNA that might exist in the genome of *Fol* by homologous comparison. However, with the existing gene annotation of *Fol*, only *FolRDR1* was identified.

Literatures have provided proof-of-concept that RNAi-based plant protection is an effective strategy for controlling crop fungal diseases. Here, we explored the potential and the mechanism of an RNAi-based crop protection strategy using direct applications of *FolRDR1*-dsRNA to inhibit *F. oxysporum*. We found that *FolRDR1* mediated the pathogen development and pathogenicity. Both *Fol* and host plant efficiently took up *FolRDR1*-dsRNA from the environment. Finally, exogenous application of *FolRDR1*-dsRNAs significantly alleviated the progress of tomato wilt disease symptoms. In summary, our data established that SIGS based on *FolRDR1*-dsRNA-RNAi contributed to the resistance to tomato Fusarium wilt disease.

## Results

### *FolRDR1* is required for the vegetative growth and asexual reproduction in *Fol*

By analysis of sequence homology, we found that *FolRDR1* (Sequence ID: XM_018384343.1) was highly conservative in *F. oxysporum* including *F. oxysporum* f. sp. *Lycopersici*, *F. oxysporum* f. sp. *Cepae*, *F. oxysporum* f. sp. *Cubense*, *F. oxysporium* f. sp. *Cucumerinum*, *F. oxysporum* f. sp. *Pisi*, *F. oxysporum* f. sp. *Raphani* and *F. oxysporum* f. sp. *Conglutinans* (S1 Fig). These results indicated that *FolRDR1* may possess essential biological functions in *F. oxysporum*. To confirm this hypothesis, we generated two *FolRDR1*-knockout (*FolRDR1*-KO) strains *FolRDR1*-KO-36# and 126# by approach of homologous recombination (S2 Fig).

To assess the potential regulated gene by impaired *FolRDR1*, the miRNA levels were evaluated by sRNA-seq using the KO-strains *FolRDR1*-KO-36 (named as *FolRDR1*-1 in library), *FolRDR1*-KO-126 (named as *FolRDR1*-2 in library) and wild type strain (named as *Fol*-WT in library). The DEGs of miRNAs were listed in S2 Table. Well correlation was showed between *FolRDR1*-KO-36 and *FolRDR1*-KO-126 (S3A Fig). Compared with WT strain, the abundances of miRNAs declined significantly in both *FolRDR1*-KO strains (S3B and S3C Fig). We further analyzed the biological functions of predicted targets of miRNAs (Listed in S3 Table), and the results showed that knockouting of *FolRDR1* mainly affected the metabolic pathway in both KO strains (S4 Fig). With above results, we concluded that the levels of miRNA were correlated with *FolRDR1* in *Fol*.

We further checked the growth rate of wild type (WT) *Fol* and *FolRDR1*-KO strains cultured on PDA plates, respectively. Statistic data showed that no significant difference was observed between *Fol* and *FolRDR1*-KO strains (S5 Fig). Further, no significant difference in colony morphology was observed either under salt, alkali and osmotic pressure stress at different concentrations (S6 Fig). The above results indicated that the absence of *FolRDR1* did not change the response to abiotic stresses in *Fol*. However, we found that the colony edge of both *FolRDR1*-KO strains were more loose than *Fol*. Furthermore, the mycelia of *FolRDR1*-KO strains presented abnormal growth such as mycelia ablation and increased sclerotia (Fig 1A). To filamentous fungi, sporulation ability is an important physiological index to measure the pathogenicity. Knocking out *FolRDR1* also leaded to lower sporulation but larger size of conidia compare to WT strain (Fig 1B–1D). Intriguingly, the growth and penetrability of *Fol* on PDA plate covered with cellophane were nearly unchanged, contrarily, both *FolRDR1*-KO strains showed dramatic decreased penetrability (Fig 1E). There results indicated that *FolRDR1* was essential to the vegetative growth and conidiogenesis in *Fol* as well as penetrability.

### *FolRDR1* is required for pathogenicity in *Fol*

To investigate the response of *FolRDR1*upto *Fol* infection in tomato roots, two-week susceptible cultivar Moneymaker seedlings were infected by *Fol*. Fusarium wilt symptoms were developed at early stage (7 day post infection, 7dpi) (Fig 2A). Total RNA was extracted from infected tomato roots which included both tomato and pathogen RNA. The transcript level of *FolRDR1*was further valuated by Northern blot, and the results indicated that *FolRDR1*was constantly induced during the pathogen infection (Fig 2B).

To further evaluate the pathogenicity of *FolRDR1*, tomato seedlings were inoculated with *Fol* and two *FolRDR1*-KO strains, respectively. We observed impaired infection of Moneymaker supported by alleviated Fusarium wilt symptoms, less presence of the fungus within the plant stem and fungal mycelium regeneration compared to *Fol*-treated Moneymaker, while no Fusarium wilt symptoms were observed in resistant cultivar Motelle under infection with all

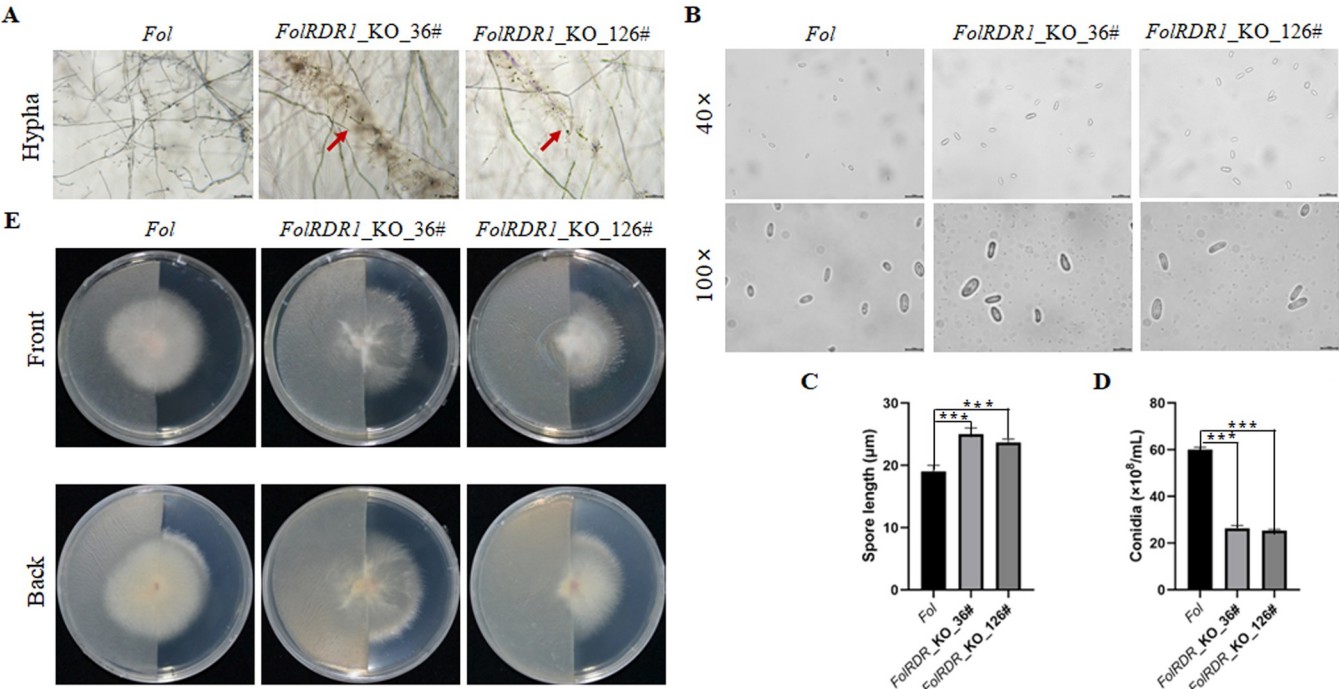

**Fig 1. Knocking out *FolRDR1* partially dampens the development of *Fol*. A** the mycelia of *FolRDR1*-KO strains presented abnormal growth such as mycelia ablation and increased sclerotia (indicated by red arrow). All three strains were cultured on PDA plates, and images were taken at fourth days. **B-D** Knocking out *FolRDR1* leaded to lower sporulation but larger size of conidia compare to WT strain. * indicates significant difference when compared to WT at $P < 0.05$, chi-square test, Error bars indicate the Standard Deviation of three replicates. 40 x scale bars, 50 μm, 100 x scale bars, 20 μm. **E** Knocking out *FolRDR1* resulted in dramatic decreased penetrability in *Fol*. All three strains were cultured on the center of PDA plates covered with half cellophane, and images were taken at fourth days. Front, images were taken from the front of plate. Back, images were taken from the back of plate. Three biological replicates were used in each experiment.

three individual strains (Fig 2C–2E). Based on above results, we concluded that FolRDR1 was a critical pathogenic factor in *Fol*.

## Environmental dsRNA-*FolRDR1* is effectively taken up by *Fol*

To validate whether SIGS of *FolRDR1* control Fusarium wilt disease, we generated *FolRDR1*-dsRNA2 (1–804 bp) and *FolRDR1*-dsRNA1 (787–1523 bp) expressing constructs under double T7 promoter. Full length *GFP*-dsRNA was constructed as a control (S7 Fig). These constructs were transformed into RNase III deficient *E. coli* HT115 (DE3) strain to express dsRNA *in vitro* under optimum condition: 0.8 mmol/L IPTG, 280 rpm/min, 37°C and 8-hour induced expression (S8 Fig).

To trace external *FolRDR1*-dsRNA, we synthesized fluorescein-labeled *FolRDR1*-dsRNA using Fluorescein-12-UTP *in vitro*. Fluorescein-12-UTP and water treatment were used as negative controls. WT conidia were cultured with fluorescein-labeled *FolRDR1*-dsRNA for 24 hours on PDA plate followed by detecting fluorescence signals. Fluorescence signals of *FolRDR1*-dsRNAs and *GFP*-dsRNA were observed in conidia (Fig 3A, left). To further confirm whether the dsRNAs enter *Fol* cells, the *Fol* mycelium were cultured with the fluorescein-labelled *FolRDR1*-dsRNAs and *GFP*-dsRNA respectively in liquid culture for 48 hours. Subsequently, conidia were collected for protoplast preparation. Similarly, fluorescence signals of *FolRDR1*-dsRNAs and *GFP*-dsRNA were observed clearly in protoplast (Fig 3A, middle). Further, we also observed the fluorescence signals in 48-hour hyphae (Fig 3A, right). Above results

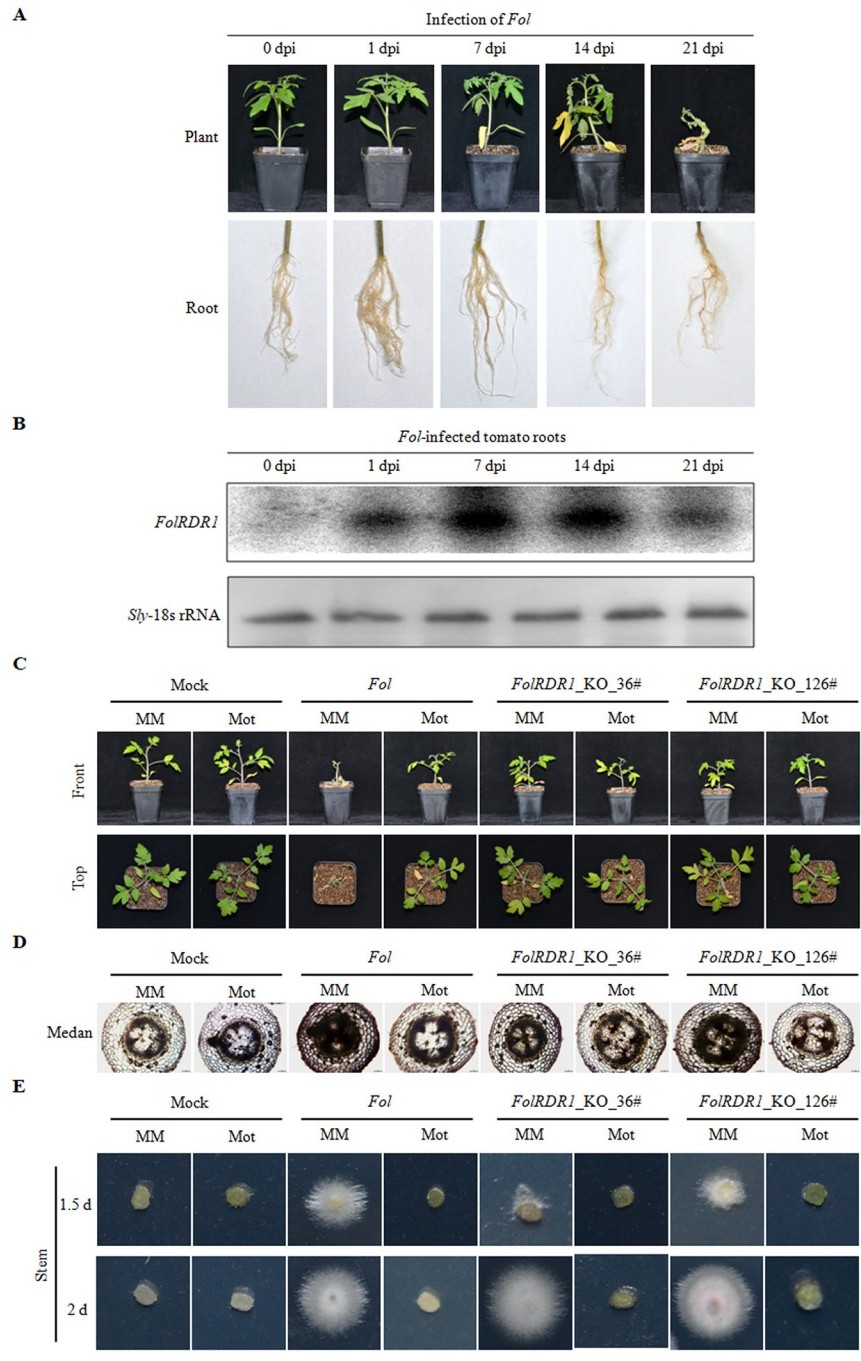

**Fig 2. *FolRDR1* is essential for pathogenicity in *Fol*. A** The wilt disease symptoms of susceptible cultivar Moneymaker infected with *Fol*. **B** The transcript level of *FolRDR1* was induced under the infection of *Fol*. 10 μg of total RNA was resolved by electrophoresis using urea polyacrylamide gel electrophoresis (PAGE) and transferred to anylon N+ membrane. [γ-32P]ATP-labelled specific nucleotide probe sequences were used for hybridization. *Sly*-18s rRNA was used as a loading control. **C** *FolRDR1* was required for pathogenicity in *Fol*. The *FolRDR1*-KO strains and control WT *Fol* were used to inoculate tomato seedlings. Wilt disease symptoms were photographed 2 weeks after inoculation. **D** Cotton blue staining results reflect the abundance of *Fol* in the stem of tomato plants. More intense cotton blue staining correlates with higher abundance of *Fol*. **E** The outgrowth of fungi from tomato stems of plants inoculated with the indicated strains on PDA, and images were taken at 1.5 and 2 days, respectively. Front, images were taken from the front of plants. Top, images were taken from the top of plants. Three biological replicates were used in each experiment.

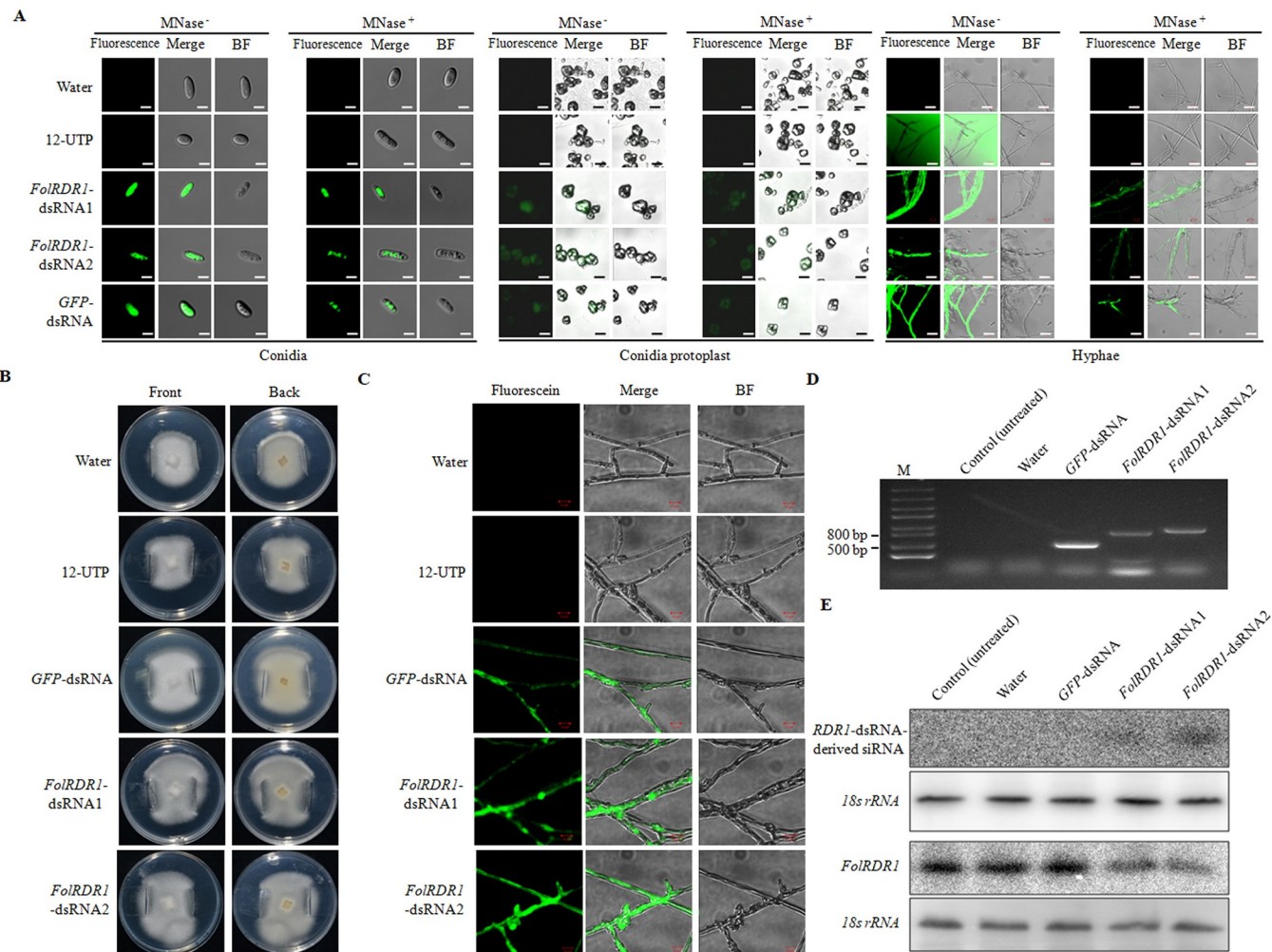

**Fig 3. *Fol* efficiently uptakes fluorescein-labelled dsRNA. A** Conidia were cultured in Vogel's minimal medium at a concentration of $10^5$ spores/mL with 150 ng/mL dsRNA for 24 hours (Left, scale bars, 10 μm), and conidia protoplast were made subsequently (Middle, scale bars, 10 μm). Hyphae were collected after 48 hours (Right, scale bars, 1 mm). Samples were treated with micrococcal nuclease (MNase) 30 min before images were taken using the confocal microscopy laser scanner (CMLS). **B** Two vertical glass slides (1 cm x 2 cm) were inserted into the PDA medium about 2 cm far away from the inoculation site of *Fol* to minimize the dissociative fluorescein-labeled dsRNA in the plate. Front, images were taken from the front of plate. Back, images were taken from the back of plate. **C** After 3 days of mycelium growth and expansion, the fluorescence signals in the mycelia climbing the glass slides were checked using CMLS. Scale bars, 1 mm. Three biological replicates were used in each experiment. **D** The corresponding dsRNA fragments in the marginal mycelia were further detected by RT-PCR. **E** *FolRDR1*-dsRNA-derived siRNAs were enriched in both *FolRDR1*-dsRNA treated fungal mycelia resulting in repressed the transcript levels of *FolRDR1*. Total ten 30-nt DNA fragments (detailed in S5 Table), which uniformly distributed in the predicted regions in the CDS of *FolRDR1*, were synthesized and mixed as a pool followed by labeling with [γ-32P]ATP as probes for to detect the enrichment of *FolRDR1*-dsRNA-derived siRNAs (Up panel). [γ-32P] ATP-labeled specific nucleotide probe sequences of *FolRDR1* were used for hybridization to detect the transcript levels of *FolRDR1* (Bottom panel). *Sly*-18s rRNA was used as a loading control in both Northern blots, respectively.

indicated that *Fol* cells took up dsRNAs from the environment, and this action is not likely to be selective.

To determine whether dsRNA can be effectively transported in *Fol* mycelium, we applied fluorescein-labeled dsRNA on the center of PDA plate inoculated with *Fol*. Two glass slides (1 cm x 2 cm) were inserted into the medium about 2 cm far away from the inoculation site to minimize the dissociative dsRNA in the plate (Fig 3B). After 3 days of mycelium expansion, we detected the fluorescence signals in the fungal mycelia climbing on the glass slides (Fig 3C). The corresponding dsRNA fragments in the fungal mycelia were further detected (Fig 3D).

To verify the production of *FolRDR1*-dsRNA-derived siRNAs, firstly, the efficient siRNAs generated in different regions of *FolRDR1* were analyzed using siRNA-Finder (Si-Fi) (http://www.wheatgenome.info/) (S9 Fig) [19]. Total ten 30-nt DNA fragments (detailed in S5 Table), which uniformly distributed in the predicted regions, were synthesized and mixed as a pool followed by labeling with [γ-32P]ATP as probes for Northern blot. The data clearly shown that *FolRDR1*-dsRNA-derived siRNAs were enriched in both *FolRDR1*-dsRNA1 and *FolRDR1*-dsRNA2 treated fungal mycelia, respectively (Fig 3E, up panel). Furthermore, the transcript level of *FolRDR1* in both *FolRDR1*-dsRNAs treated fungal mycelia were repressed significantly compared to *Fol* or water treatment samples (Fig 3E, bottom panel). These results indicated that external dsRNAs were effectively transported in *Fol* growing-mycelium and repressed the transcript level of *FolRDR1*.

## Application of external *FolRDR1*-dsRNA destructs the biological functions of *FolRDR1* in *Fol*

Previously, we have shown that *FolRDR1* was essential to the vegetative growth and conidiogenesis in *Fol*. We further explore whether external *FolRDR1*-dsRNA impair the biological functions of *FolRDR1* in *Fol*. We applied *FolRDR1*-dsRNA on the center of the plate colony and observed the colony growth at 5 dpi. The data showed that the growth rate of the mycelium were unchanged under the treatments of *FolRDR1*-dsRNAs and *GFP*-dsRNA (Fig 4A and 4B). To the conidiogenesis, however, both *FolRDR1*-dsRNA1 and *FolRDR1*-dsRNA2, but not *GFP*-dsRNA, significantly suppressed the production of conidia (Fig 4C). Moreover, the transcript level of *FolRDR1* was repressed accordingly at optimal treatment with the concentration of 150 ng/mL and 24 hours in liquid PDA medium (Fig 4D and 4E). We also noticed that the RNAi-based silencing efficiency of two dsRNAs in interference processing was about 90% for *FolRDR1*-dsRNA1 and 81% for *FolRDR1*-dsRNA2, respectively (Fig 4D and 4E), which was partially addressed by the analysis using siRNA-Finder (Si-Fi) (S9 Fig). Above results indicated that *FolRDR1*-dsRNA1 generated efficient siRNAs more than *FolRDR1-dsRNA2*, and application of external *FolRDR1*-dsRNAs effectively destructed the biological functions of *FolRDR1* in *Fol*.

## Host plant takes up and transfers environmental *FolRDR1*-dsRNAs

To evaluate the residual period of the external dsRNA on the host leaves, *FolRDR1*-dsRNAs and *GFP*-dsRNA were daubed on the 2-week tomato seedling leaves. The treated leaves, stems and roots were collected at different time points for detecting dsRNA by RT-PCR. All three dsRNAs were detected until 7 dps (day post spray) which indicated that the environmental dsRNA could remain on leaves, stems and roots for at least 7 days without dsRNA selectivity (Fig 5A).

To verify whether external dsRNA applied on tomato leaf may be effectively transferred in plant tissues, one side leaf was daubed with fluorescein-labeled dsRNA followed by checking the fluoresce signals in undaubed leaf, root and stem. No fluoresce signal was detected in water treated leaf which indicated no background excitation fluorescence in nature tomato leaf. On the other hand, strong fluoresce signals were presented in daubed leaf (Fig 5B). Then, visible fluoresce signals were detected in undaubed leaf 2 days after treating for all three dsRNAs (Fig 5C). Intriguingly, relatively strong fluoresce signals were shown in stem and root compared to leaf. In stem, fluoresce signals were obviously emerged in vascular bundle (Fig 5D), and distributed in the entire root, especially in root hair (Fig 5E). Above results demonstrated that tomato plant effectively took up the external *FolRDR1*-dsRNAs and transferred to the different tissues.

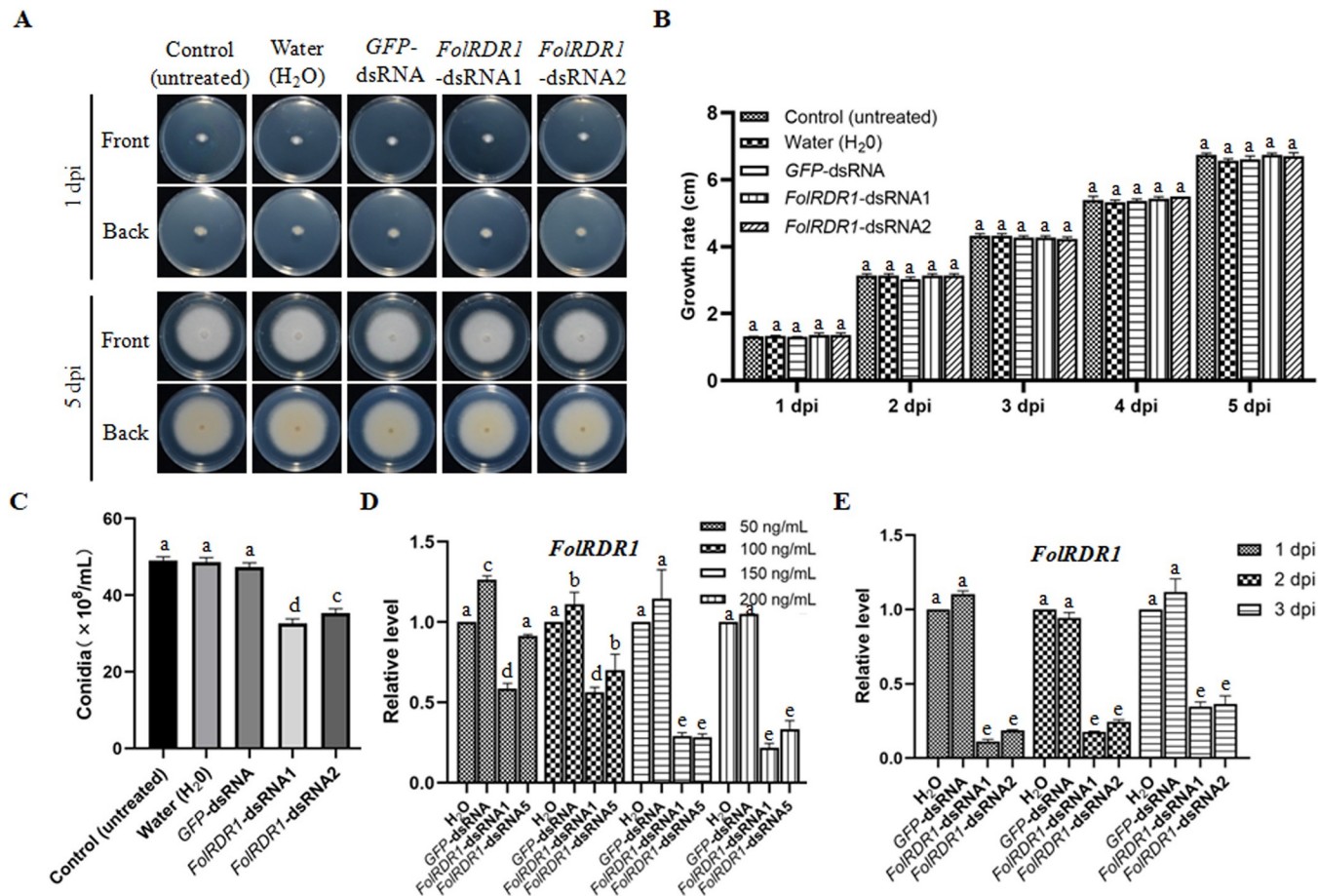

**Fig 4. Application of exogenous dsRNA suppresses the production of conidia and expression level of *FolRDR1* in *Fol*. A** The growth rate of the mycelium were unchanged under the treatments of either *FolRDR1*-dsRNAs or *GFP*-dsRNA at a concentration of 150 ng/mL. *Fol* strain was cultured on PDA plates with the treatments of either *FolRDR1*-dsRNAs or *GFP*-dsRNA, respectively, and images were taken at 1 day and 5 days. Front, images were taken from the front of plate. Back, images were taken from the back of plate. **B** The growth rate of the mycelium was scaled at different time points. **C** The treatments of *FolRDR1*-dsRNAs suppressed the production of conidia. **D** The transcript levels of *FolRDR1* in both *FolRDR1*-dsRNAs treated marginal mycelia were concentration dependent. **E** The transcript level of *FolRDR1* in both *FolRDR1*-dsRNAs treated marginal mycelia were time dependent. Three biological replicates were used in each experiment. a presents no significant differences (p>0.05), c, d, e present significant differences (P<0.01).

## External application of *FolRDR1*-dsRNA alleviates the development of tomato wilt disease

To assess whether SIGS attenuate *Fol* infection, we sprayed the *Fol* pre-treated 2-week tomato seedling with *FolRDR1*-dsRNAs (200 ng/mL) at 24 hours after infecting with *Fol* and scaled the development of Fusarium wilt symptoms. At 15 dpi without spraying treatment, susceptible cultivar Moneymakers showed the initial symptoms of tomato wilt disease with cotyledon chlorosis and wilting. Meanwhile, Moneymakers treated with *FolRDR1*-dsRNA1 and *FolRDR1*-dsRNA2, respectively, developed severer Fusarium wilt symptoms with euphylla chlorosis and wilting. However, at 25 dpi, Moneymakers treated with or without *GFP*-dsRNA as negative controls were gradually died, showing severe symptoms of Fusarium wilt. However, the wilt disease symptoms of Moneymakers treated with *FolRDR1*-dsRNA1 and *FolRDR1*-dsRNA2 were significantly alleviated (Fig 6A). By staining for the presence of the fungus within the plant stem and fungal mycelium regeneration, we further observed alleviated infection in Moneymaker treated with *FolRDR1*-dsRNA1 and *FolRDR1*-dsRNA2, respectively,

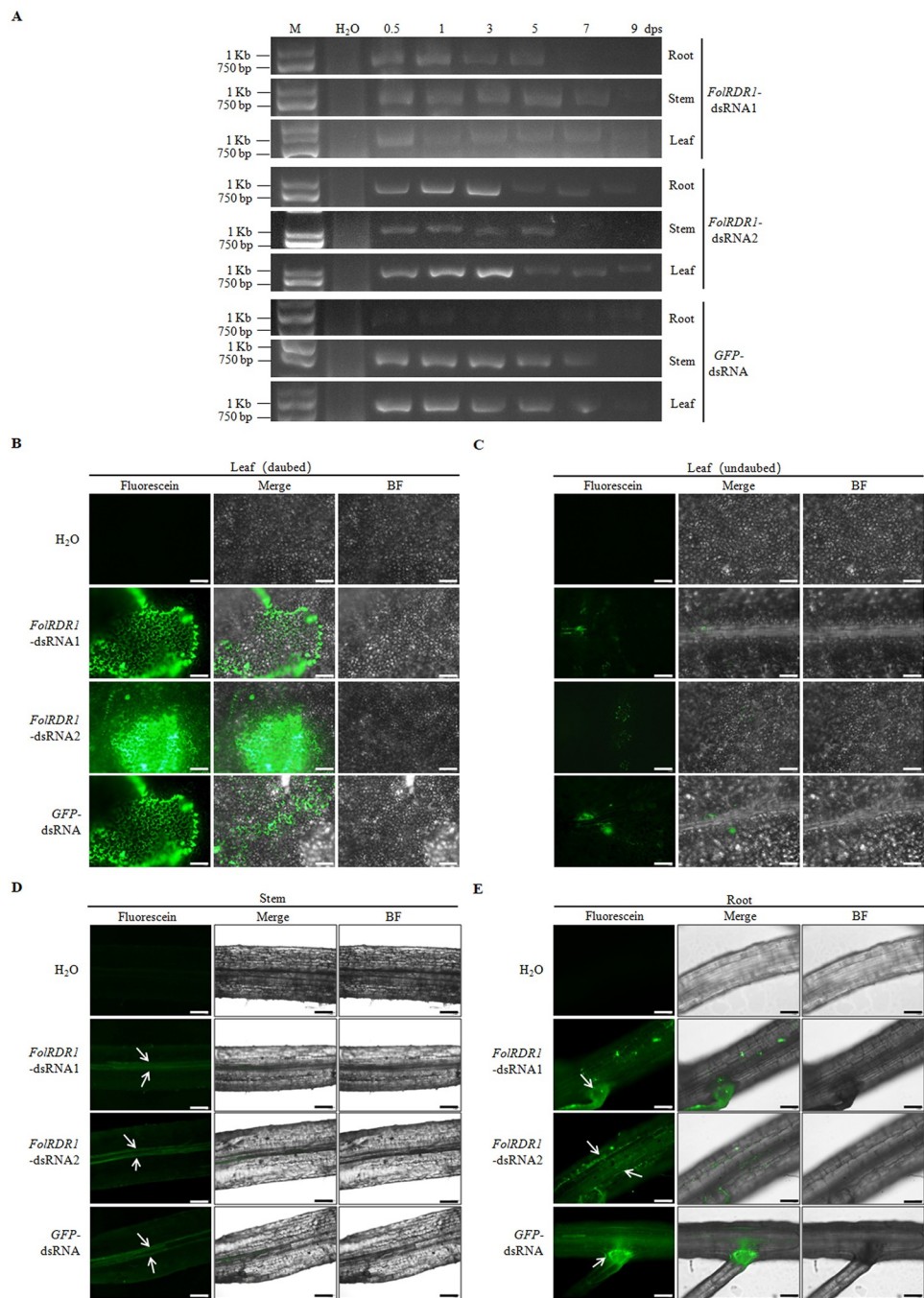

**Fig 5. External dsRNA was transported in plant tissues. A** *FolRDR1*-dsRNAs and *GFP*-dsRNA were sprayed on the 2-week tomato seedling leaves. The different tissues of treated plant including leaf, stem and root were collected at different time points for detecting dsRNA by RT-PCR. **B** Fluorescein-labeled dsRNAs were daubed on one side leaf followed by checking the fluoresce signals using CMLS after 24 hours. Scale bars, 1 mm. **C** The fluoresce signals on undaubed leaf were detected using CMLS 3 days after treatment described above. Scale bars, 1 mm. **D** The fluoresce signals in stem were detected using CMLS 3 days after treatment described above. Vascular bundles were pointed by white arrows. Scale bars, 1 mm. **E** The fluoresce signals in root were detected using CMLS 3 days after treatment described above. Vascular bundles were pointed by white arrows. Scale bars, 1 mm.

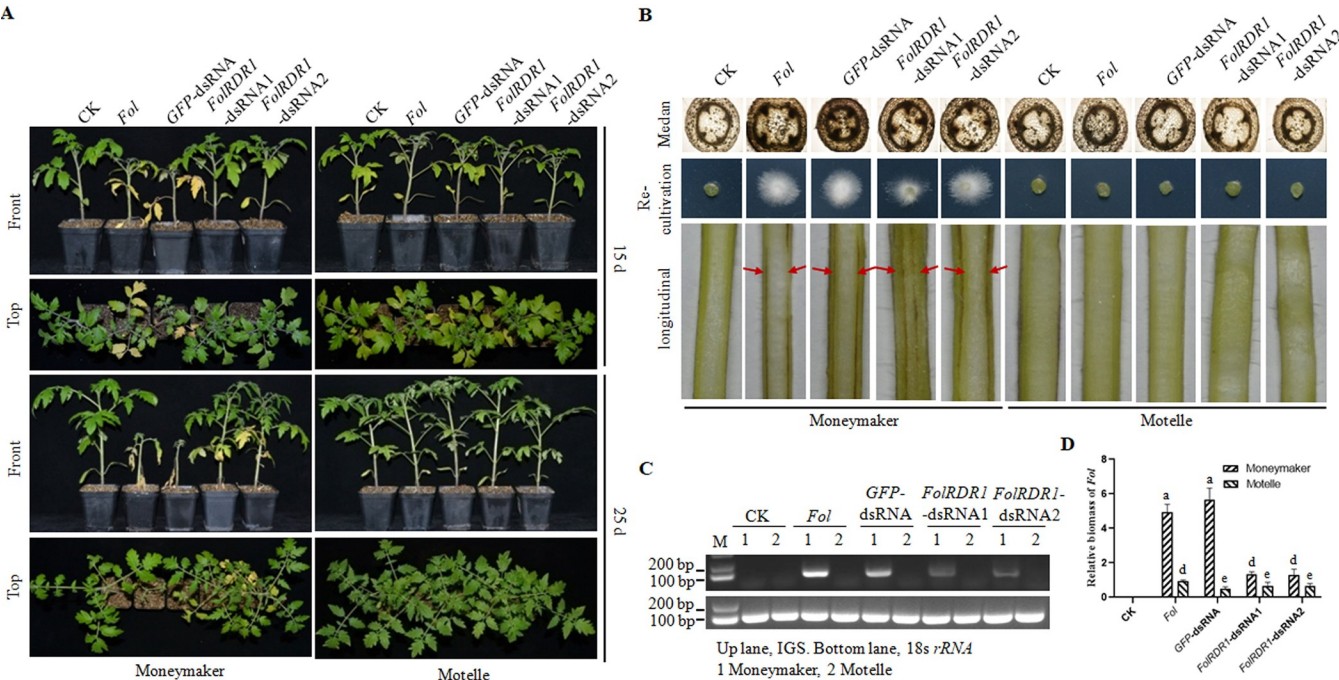

**Fig 6. Exogenous application of *FolRDR1*-dsRNA alleviates the development of Fusarium wilt disease in *Fol* pre-treatment tomato seedlings. A** Two-week tomato seedlings were infected by WT *Fol* as described previously, followed by spraying *FolRDR1*-dsRNAs and *GFP*-dsRNA on the leaves, respectively. Wilt disease symptoms were photographed 2 weeks after inoculation. Front, images were taken from the front of plants. Top, images were taken from the top of plants. **B** Cotton blue staining results reflect the abundance of *Fol* in the stem of tomato plants. More intense cotton blue staining correlates with higher levels of *Fol* (Up panel). The outgrowth of fungi from tomato stems of plants inoculated with the indicated strains on PDA, and images were taken at 2 dpi, respectively (Middle panel). Diseased vascular bundles were checked in longitudinal splitting stem (Pointed by red arrows) (Bottom panel). **C** Biomass of *Fol* in according treated plants was detected by PCR. **D** Biomass of *Fol* in according treated plants was detected by qPCR. Three biological replicates were used in each experiment. a presents no significant differences (p>0.05),the largest average, d, e present significant differences (P<0.01).

as well as less presence of the fungus within the plant stem and fungal mycelium regeneration compared to *Fol*-treated Moneymaker, while no Fusarium wilt symptoms were observed in resistant cultivar Motelle under infection with all three individual strains (Fig 6B). The results of relative biomass of *Fol* in different sample further supported above impaired infection between different treatments which correlated with the symptoms of Fusarium wilt (Fig 6C and 6D).

Actually, we performed both pre-treatment (Fig 6) and after-treatment (S11 Fig) with *FolRDR1*-dsRNAs experiments. The data indicated that no significant difference was shown between these two treatments. Due to the unpredictable development of fusarium wilt disease in field, we think that application in the early stage of disease might reduce the cost of disease control. Based on these results, we concluded that SIGS of *FolRDR1*-dsRNAs attenuated tomato Fusarium wilt disease under lab conditions.

To validate the potential off-target of *FolRDR1*-dsRNAi, the off-targets were searched and predicted in different species using si-Fi algorithm with splitting *FolRDR1*-dsRNAi trigger sequence (the complement to the target sequence of the corresponding RNA) into all possible MERs [20]. Our data showed that no off-target of *FolRDR1*-RNAi in *Fusarium oxysporum* was predicted, and no off-target was predicted in *Solanum lycopersicum* (tomato), either (S4 Table). We further generated RNA-seq libraries using water (Mock), *GFP*-dsRNA (negative control), *FolRDR1*-dsRNA1 treated tomato seedlings to analyze the transcriptom of tomato. The data indicated that no significant changes were found between SIGS-*FolRDR1* and two controls (S10 Fig, and all DEGs of mRNA were listed in S6 Table) (The raw sequence data for

this study are available in the National Genomics Data Center with accession no. CRA011174, https://bigd.big.ac.cn/gsa/browse/CRA011174). Taken together, our results supported that *FolRDR1* could be considered as a suitable candidate for developing biological agent to control tomato wilt disease.

## Discussion

Recently, studies illustrated that spraying dsRNAs/sRNAs targeting essential pathogen genes on plant surfaces afforded efficient crop protection, and RNAi-based SIGS strategy of disease control was potentially sustainable eco-friendly alternative to standard chemical pesticides for controlling agricultural losses caused by pests and pathogens [7,21]. Exogenous dsRNA triggering suppression of gene activity in a homology-dependent manner was firstly discovered in *Caenorhabditis elegans* [22]. Since then, SIGS was known as a powerful, fast, and environmentally friendly strategy to circumvent the problems in creating GMOs [7,13,23,24].

RDR1 (also termed as Rdp1) localizes to all known heterochromatic loci and is required for sense transgene-induced silencing to generate dsRNA molecules in fungi [25,26]. RDRs have mainly been described to be involved in amplification of RNAi in eukaryotes [18]. In the present study, we found that *FolRDR1* mediated the invasion to the host plant tomato, and played as an essential regulator in pathogen development and pathogenicity in *Fol*. However, no similar biological functions of *RDR1* were reported in other fungal so far. Even more intriguing, we found that abolishing of *FolRDR1* leaded to mycelia ablation and abnormal sclerotia in *Fol* (Fig 1A), which highlights the importance to study underlying mechanisms.

Host *Arabidopsis* cells secreted exosome-like extracellular vesicles to deliver sRNAs into fungal pathogen *B. cinerea*, and transferred host sRNAs induced silencing of fungal genes critical for pathogenicity [27]. SIGS has been shown to be effective on controlling plant disease initialed by taking up external dsRNA. The uptake efficiency of dsRNA was significantly different among various fungi with exception such as *Colletotrichum gloeosporioides* [13]. Our present results showed that both fungal pathogen *Fol* and host plant could uptake *FolRDR1/GFP*-dsRNAs directly from environment without obvious selectivity (Fig 3), and dsRNAs were transferred in different tissues efficiently (Fig 4). These results promote us speculate that fungal pathogens take up external RNA unselectively but species dependently.

More than intriguing, the fluorescence signals of *FolRDR1* /GFP-dsRNAs were dominantly localized in the host plant vascular bundles (Fig 5D and 5E). Germination of dormant spores in soil results in adherence and invasion of plant roots by *Fol* hypha, subsequently, move from the root cortex to the vascular bundles where microconidia spore are produced and disseminated. Using the vascular bundles as transport corridor, endogenetic hypha spreading to aboveground tissues is critical for disease progression for *Fol*. The characteristic wilt symptoms appear as a result of severe water stress, mainly due to vessel clogging [28]. Further questions need to be answered how these external dsRNAs are taken up by plant and fungal cells directly from environment traveling across the boundaries between organisms of different taxonomic kingdoms.

Previously, *Arabidopsis* and barley ectopically expressing a double-stranded RNA (dsRNA) targeting three fungal *CYP51* genes significantly enhanced plant resistance to *Fusarium graminearum* species by disrupting fungal membrane integrity, subsequently, spraying detached barley leaves with a 791-nt long CYP3-dsRNAs that contains complementary sequences to *CYP51*family members prior to fungal infection could effectively inhibit disease and yield much smaller lesions [14,29]. Similarly, externally applying dsRNAs and small RNAs (sRNAs) targeting Dicer-like protein genes DCL1 and DCL2 of *B. cinerea* on vegetables, fruits, and flower petals could suppress grey mold disease effectively [12]. Here, applying *FolRDR1*-

dsRNAs that target *FolRDR1* on the surface of pre-infected tomato seedling leaves significantly inhibited the development of Fusarium wilt (Fig 6). To develop a successful SIGS-based crop protection strategy, several critical aspects must be considered. Firstly, a reasonable duration of efficacy is desired. Our data revealed that the *FolRDR1/GFP*-dsRNAs could be detected even at 7 dps of the local sprayed site, suggesting either external RNAs were stable for at least seven days on the surface of the leaves and/or remained stable in the plant cells (Fig 5A). Secondly, off-target is another considered factor for eco-friendly alternative to standard chemical pesticides. By bioinformatics prediction, our data showed that *FolRDR1*-RNAi resulted in no target-specific either in fungal pathogen or host plant (S2 Table, S8 Fig).

Taken together, eukaryotic pathogens, including fungi and oomycetes, cause vast worldwide economic losses in crop annually. Compared to traditional chemical pesticides, our collective data provided solid evidences that *FolRDR1*-RNAi-SIGS is an advantageous artificial trans-kingdom RNAi-based bio-pesticide to protect tomato from Fusarium wilt disease. Application strategies can be improved by encapsulating with chemical reagents to stabilize the dsRNAs and thus increase the strength and duration of plant protection. Such specific pathogen gene-targeting RNAs represents a new generation of environmentally-friendly fungicides for increasing safety and quality of crop yields to feed the growing population.

## Materials and methods

### Plant materials, fungal inoculation, measurements of *Fol* biomass

Two previously described tomato near-isogenic cultivars, susceptible Moneymaker (MM, *i2/i2*) and resistant Motelle (Mot, *I2/I2*), were employed in this study [30,31,32]. Briefly, tomato seedlings were grown in long-day conditions (16 hr light/8 hr dark, at 25˚C, 65% humidity, photon flux density 40 μmol m$^{-2}$ s$^{-1}$) for 2 weeks for pathogen inoculation.

The pathogenic fungal strain is *Fusarium oxysporum* f. sp *lycopersici* (race 2, FGSC 9935, *Fol*). *Fol* was grown on potato dextrose agar medium (PDA) for 5 days at 28˚C with constant light. Spore suspensions were prepared by harvesting cultures in Vogel's minimal medium at a concentration of 10$^7$ spores/mL. Tomato seedlings were removed from soil, and rinsed with tap water roots were inoculated with *Fol* spores for 30 min. Water treatment was used as a mock control. All experiments were conducted using three biological replicates.

To assess the relative levels of *Fol* biomass in tomato tissues, genomic DNA was isolated from tomato tissues using CTAB [33]. The rDNA Intergenic Spacer Region (IGS) of *Fol* was amplified from genomic DNA using qPCR (Primers listed in S1 Table) as a marker to assess relative fungal biomass [2].

### Statistics of spore production

All strains were inoculated on PDA medium for 4 days. 5 mm mycelium piece at the edge of the colony was cut and cultivated in 100 mL of Vogel's liquid medium at 28˚C, 200 rpm for 4 days. Spores were collected using three layers of sterile gauze. After spinning down, the number of spores were counted using hemocytometer under the optical microscope. These experiments were repeated three times and three biological replicates in each experiment.

### Determination of fungal penetrability

In this experiment, cellophane was used to mock plant cell wall to test the fungal penetrability. The cellophane was cut to a semicircle piece with a radius of 4 cm and placed on the PDA medium plate. 5 mm mycelium piece at the edge of the colony was cut and cultivated in the center of the plate under light at 28˚C for 5 days. The colony morphology and mycelium

morphology were recorded. These experiments were repeated three times and three biological replicates in each experiment.

## Total RNA extraction, Northern blotting, quantitative real-time PCR (qRT-PCR)

Total RNA was extracted using the Trizol reagent (#15596026, Invitrogen, CA, USA). Purified RNA was treated with DNase I (Thermo Fisher Scientific, Waltham, Ma, USA). For small RNA gel blots, 40 μg of total RNA was separated on 7 M urea 15% denaturing polyacrylamide gels in Tris/Boric Acid/EDTA (1X TBE), followed by transferring to a nylon N$^+$ membrane. For high molecular weight RNA gel blots, 10 μg of total RNA was resolved by electrophoresis using urea polyacrylamide gel electrophoresis (PAGE) and transferred to anylon N$^+$ membrane. Gene-specific nucleotide probes (Primers listed in S1 Table, and DNA fragments listed in S5 Table). Gene-specific nucleotide probes (Primers listed in S1 Table) were end-labeled using [γ-32P]ATP (#M0201, New England Biolabs, Ipswich, MA; nucleotide probes were labeled according to the manufacturer's recommendations). Blots were stripped and re-probed using a *Sly*-18s rRNA nucleotide probe to provide a loading control. All blots were imaged using a PhosphorImager (Molecular Dynamics/GE Life Sciences, Pittsburgh, PA) [30].

For reverse transcriptase-polymerase chain reaction (RT-PCR), first-strand cDNA was synthesized from 1 μg of total RNA using the Superscript III First-Strand Synthesis System (#18080051, Thermo Fisher Scientific, Waltham, Ma, USA) according to the manufacturer's recommendations (Primers listed in S1 Table). Diluted cDNA was used as the template for quantitative RT-PCR (#1708880, Bio-Rad, Philadelphia, PA, USA), using *Sly*-18s rRNA as the internal control. Differential expression of genes was calculated using the $2^{-\Delta\Delta Ct}$ method [34].

## Construction of *FolRDR*1 knockout strains

*FolRDR*1 knockout mutant strains were generated by using the split-marker approach previously described by our laboratory [35]. Briefly, for *FolRDR*1 knockout vector construction, the upstream flanking sequence, downstream flanking sequence of *FolRDR*1 and *HPH* cassette were amplified and purified, followed by transformation into protoplasts of the wild-type strain (Primers listed in S1 Table). The knockout construct was then transformed into protoplasts of *Fol*. Transformants with the desired genetic changes were identified using site-specific primer pairs (Primers listed in S1 Table).

## Synthesis of dsRNA and uptake of fluorescein-labelled dsRNA *in vitro*

Synthesis of dsRNA *in vitro* was based on established protocols [13]. Briefly, selected fragments of *FolRDR*1 were amplified using gene-specific primers and inserted into the pL4440 vector containing double T7 promoter (Primers listed in S1 Table). *FolRDR*1/*GFP*-dsRNA was labeled using the fluorescein RNA Labeling Mix Kit following the manufacturer's instructions (#11685619910, Sigma, St. Louis, MO, USA). For confocal microscopy examination of fluorescein-labelled dsRNA uptake by fungal mycelium, 5 μL of 150 ng/μL fluorescent dsRNA was applied to the PDA medium or the microscope slides surface.

## Light microscopy studies

To track the fluorescein-labeled *FolRDR1*/*GFP*-dsRNA, plant tissues and fungal materials were collected after dsRNA treatment with the concentration of 200 ng/μL and 150 ng/mL, respectively. Images were taken using a Zeiss LSM 710 confocal microscope with a 63/1.2 NA

C-Apochromat oil immersion objective (Zeiss, Oberkochen, Germany). The relative fluorescent density was analyzed usingImage-pro Plus (Media Cybernetics Inc., Shanghai, China).

## Construction of sRNA-seq and RNA-seq libraries and analysis

For sRNA-seq, total RNA of the KO-strains *FolRDR1*-KO-36 (named as *FolRDR1*-1 in library), *FolRDR1*-KO-126 (named as *FolRDR1*-2 in library) and wild type strain (named as *Fol*-WT in library) were extracted individually using the TRIzol reagent (#15596026; Life Technologies) according to the manufacturer's recommendations. For each Illumina library, 1 μg total RNA was used, according to the manufacturer's instructions. The libraries were subsequently sequenced using the Illumina HiSeq 2000 (Biomarker Technologies, Rohnert Park, CA, USA). For RNA-seq, two-week-old tomato seedlings were pre-infected with *Fol* followed by spraying *FolRDR1/GFP*-dsRNA with the concentration oxcf 200 ng/μL. Three biological replicates were used, with 5 seedlings for each treatment. The leaves were collected and then frozen immediately in liquid nitrogen. Total RNA was extracted described previously.

For individual Illumina library, raw reads were subjected to quality control (QC). After QC, raw reads were filtered into clean reads. All sequence reads were trimmed to remove the low-quality sequences. The sequence data were subsequently processed using in-house software tool SeqQC V2.2. House-keeping small RNAs including rRNAs, tRNAs, snRNAs and snoRNAs were removed by blasting against GenBank (http://www.ncbi.nih.gov/Genbank) servers. The trimmed reads were then aligned to the *Fusarium oxysporum* and *Solanum lycopersicum* reference genome respectively using TopHat v2.0.0 and Bowtie v0.12.5 [36] with default settings. The expression levels of miRNAs or mRNAs were normalized to the reads per million (rpm) value for each individual library.

## Statistical analysis

Each result was presented as the mean ± standard deviation (SD) of at least three replicate measurements. Significant differences between treatments were statistically evaluated by SD and one-way analysis of variance (ANOVA) using SPSS 2.0 (Chicago, IL, USA). The data for two specific different treatments were compared statistically by ANOVA, followed by Student's T-test if the ANOVA result was significant at $p < 0.01$.

## Supporting information

**S1 Fig. Amino acid sequence of RDR1 alignment and phylogenetic tree construction among different *Fusarium oxysporum* races. A** Alignment of amino acid sequence of RDR1 using Pairwise Align Protein. All amino acid sequence of RDR1 were from https://www.ncbi.nlm.nih.gov. **B** The phylogenetic tree was constructed using MEGA (Molecular Evolutionary Genetics Analysis).
(TIF)

**S2 Fig. Construction of *FolRDR1* knockout strains. A** Concise schematic diagram of homologous recombination. **B** PCR fragments used for homologous recombination. **C** Diagnostic PCR was used to identify positive clones.
(TIF)

**S3 Fig. The analysis of sRNA-seq using FolRDR1-KO and WT strains.** In the library, KO-strains *FolRDR1*-KO-36 was named as *FolRDR1*-1, FolRDR1-KO-126 was named as *FolRDR1*-2, and wild type strain was named as *Fol*-WT. **A** Correlation heat map analysis. **B, C** The abundances of miRNAs declined in both FolRDR1-KO strains.
(TIF)

**S4 Fig. The analysis of GO.** Knockouting of *FolRDR1* mainly affected the metabolic pathway in both KO strains.
(TIF)

**S5 Fig. Knockouting *FolRDR1* has no effect on the growth of *Fol*. A** All strains were culture on PDA plate, and photographed at different time points. **B** The growth curve was generated based on the colony diameter. Front, images were taken from the front of plate. Back, images were taken from the back of plate. Three biological replicates were used in each experiment.
(TIF)

**S6 Fig. Knockouting *FolRDR1* has no effect on the response to abiotic stress. A** All strains were culture on PDA plate with different concentration of NaCl (Left). The growth of colony was scaled at different time points, and the growth curve was generated (Right). **B** All strains were culture on PDA plate with different pH (Left). The growth of colony was scaled at different time points, and the growth curve was generated (Right). **C** All strains were culture on PDA plate with different concentration of sorbital (Left). The growth of colony was scaled at different time points, and the growth curve was generated (Right). Front, images were taken from the front of plate. Back, images were taken from the back of plate. Three biological replicates were used in each experiment. a presents no significant differences ($p > 0.05$).
(TIF)

**S7 Fig. Construction of *FolRDR1*-dsRNA expression pL4440 vector containing double T7 promoter. A** Sketch map of *FolRDR1*-dsRNA1 and *FolRDR1*-dsRNA2. **B** Fragments of *FolRDR1*-dsRNA1, *FolRDR1*-dsRNA2 and *GFP*-dsRNA were amplified using gene-specific primers. **C** Diagnostic PCR was used to identify positive clones.
(TIF)

**S8 Fig. Establishment of heterologous expression dsRNA in *E. coli*. A** Expression strains of *FolRDR1*-dsRNA1, *FolRDR1*-dsRNA2 and *GFP*-dsRNA were induced using different concentration of IPTG. **B** Abundance of *FolRDR1*-dsRNA1, *FolRDR1*-dsRNA2 and *GFP*-dsRNA were scaled under different temperature. **C** Abundance of *FolRDR1*-dsRNA1, *FolRDR1*-dsRNA2 and *GFP*-dsRNA were scaled under different induced time points. **D** The growth curve of different strains measured by light transmittance (OD = 600 nm). a presents no significant differences ($p > 0.05$), d, e present significant differences ($P < 0.01$).
(TIF)

**S9 Fig. Prediction of efficient siRNAs generated in different regions of *FolRDR1*.** Efficient siRNAs generated in different regions of *FolRDR1* were predicted using siRNA-Finder (Si-Fi). Briefly, the off-target searching pipeline starts with splitting a long RNAi trigger sequence (the complement to the target sequence of the corresponding RNA) into all possible MERs using stringent parameters (stricter strand selection rules plus target site accessibility calculations).
(TIF)

**S10 Fig. Analysis of RNA-seq libraries using water (Mock), *GFP*-dsRNA (negative control), *FolRDR1*-dsRNA1 treated tomato seedlings.** The samples were collected at 24 hours after treatment. **A** The number of DEGs (Different Expressed Gene) in RNA-seq libraries using water (Mock), *GFP*-dsRNA (negative control), *FolRDR1*-dsRNA1. **B** Analysis of KEGG (Kyoto Encyclopedia of Genes and Genomes).
(TIF)

**S11 Fig. Exogenous application of *FolRDR1*-dsRNA alleviates the development of Fusarium wilt disease in *Fol* after-treatment tomato seedlings. A** Two-week tomato seedlings

were sprayed with *FolRDR1*-dsRNAs on the leaves respectively, followed by infecting by WT *Fol* two days later as described previously. Wilt disease symptoms were photographed 2 weeks after inoculation. Front, images were taken from the front of plants. Top, images were taken from the top of plants. **B** Cotton blue staining results reflect the abundance of *Fol* in the stem of tomato plants. More intense cotton blue staining correlates with higher levels of *Fol* (Up panel). The outgrowth of fungi from tomato stems of plants inoculated with the indicated strains on PDA, and images were taken at 2 dpi, respectively (Middle panel). Diseased vascular bundles were checked in longitudinal splitting stem (Pointed by red arrows) (Bottom panel). (TIF)

**S1 Table. Primers used in this study.**
(DOC)

**S2 Table. List of all DEGs miRNA.**
(XLSX)

**S3 Table. List of all predicted targets of miRNA.**
(XLSX)

**S4 Table. Prediction of *FolRDR1* off-target transcripts.**
(DOC)

**S5 Table. Sequences of 30-nt DNA fragments which uniformly distributed in the predicted regions in the CDS of *FolRDR1*.**
(DOC)

**S6 Table. The list of all expressed genes.**
(XLS)

## Acknowledgments

We are grateful for the gift of tomato cultivars from Dr. Isgouhi Kaloshian at the University of California, Riverside, US. We appreciate the valuable discussions with Prof. Xiao-Ming Zhang from the Institute of Zoology, Chinese Academy of Sciences. We thank Mr. Sijian Li for cooperation.

## Author Contributions

**Conceptualization:** Shou-Qiang Ouyang.

**Data curation:** Tao Feng.

**Investigation:** Hui-Min Ji, Tao Feng, Lu Cheng, Nan Wang.

**Methodology:** Hui-Min Ji, Nan Wang.

**Project administration:** Shou-Qiang Ouyang, Shu-Jie Luo.

**Software:** Shu-Jie Luo, Lu Cheng.

**Supervision:** Shou-Qiang Ouyang.

**Writing – original draft:** Shou-Qiang Ouyang.

**Writing – review & editing:** Shou-Qiang Ouyang.

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
