## [Decision Letter · Decision Letter 0]

4 Feb 2023

Dear Prof. Ouyang,

Thank you very much for submitting your manuscript "Artificial trans‐kingdom RNAi of FolRDR1 is a potential strategy to control tomato wilt disease" for consideration at PLOS Pathogens. As with all papers reviewed by the journal, your manuscript was reviewed by members of the editorial board and by several independent reviewers. In light of the reviews (below this email), we would like to invite the resubmission of a significantly-revised version that takes into account the reviewers' comments.

We cannot make any decision about publication until we have seen the revised manuscript and your response to the reviewers' comments. Your revised manuscript is also likely to be sent to reviewers for further evaluation.

Sincerely,

Nian Wang

Academic Editor

PLOS Pathogens

Shou-Wei Ding

Section Editor

PLOS Pathogens

Kasturi Haldar

Editor-in-Chief

PLOS Pathogens

orcid.org/0000-0001-5065-158X

Michael Malim

Editor-in-Chief

PLOS Pathogens

orcid.org/0000-0002-7699-2064

Reviewer's Responses to Questions

**Part I - Summary**

Reviewer #1: In the current manuscript, Ouyang and colleagues showed that RDR1 of the plant-pathogen Fusarium oxysporum is required for penetrability, sporulation and virulence on tomato. Further on, the authors utilized SIGS to silence rdr1 and demonstrated that silenced fungi display similar phenotypes as the rdr1 KO strains. Additionally, the authors used RT-PCR and labeled –dsRNA to show that RDR1-directed dsRNA is stable and can be systemically transported in fungal hyphae and the tomato plant host. Last, the authors showed that spraying tomato seedling with RDR1-directed dsRNA inhibited fungal infection and suggested that such treatment might be utilized to control fusarium wilt in the future.

The described experiments are well designed, the manuscript is well written and the results provided in this manuscript will appeal to the readers of PLOS Pathogens. However, I do have some comments about specific experiments that should be addressed.

Reviewer #2: Tomato wilt disease caused by Fusarium oxysporum f.sp. Lycopersici (Fol) is a severe fungal disease that threatens global tomato production. In the paper, they first characterize the FolRDR1 (RNA-dependent RNA polymerase 1) gene and find that it is essential for the development and pathogenicity of Fol. In this manuscript, a biocontrol agent FolRDR1-RNAi-SIGS was developed and can alleviate the development of tomato wilt disease in the lab without off-target of FolRDR1-RNAi in tomato, which is significant progress for the biocontrol of Fusarium wilt disease. The data support the claims in the paper. The manuscript is well-organized and written. I only have one major comment.

Reviewer #3: This study characterized the function of FolRDR1 (RNA-dependent RNA polymerase 1) in Fusarium oxysporum development and pathogenicity, and applied Spray-Induced Gene Silencing (SIGS) strategy to control the tomato wilt disease. The authors successfully adopted SIGS strategy in the tomato- F. oxysporum pathosystem, however, it’s not novel enough as the SIGS strategy has been successfully used in multiple pathosystems including the barley-F. graminearum pathosystem. The title of the manuscript is “Artificial trans‐kingdom RNAi of FolRDR1 is a potential strategy to control tomato wilt disease”, however, the experiments were about the taken up of external dsRNAs by Fol or tomato to induce RNAi, directly taking up of dsRNA from tomato by Fol has not been experimentally demonstrated. Although this study has the potential to provide new insight into the management of tomato Fusarium wilt disease, the manuscript in the current format lacks important information and requires extensive modifications before a new submission.

**Part II – Major Issues: Key Experiments Required for Acceptance**

Reviewer #1: -All main figures – all pictures are in poor quality and it's really difficult to judge the plate and plant phenotypes since it is all blurry.

-The choice of RDR1 as the silencing target in Fol is very peculiar and not well explained. Why would the authors want to silence a component of the cell silencing machinery? Wouldn't it hurt the efficiency of the silencing itself? Why was this gene silenced and not other housekeeping gene that is required for survival? In addition, the characterization of the rdr1 mutants appear to be detached from the putative cellular function of this gene. It will be more appropriate to check whether gene silencing is affected in this mutant as well.

-Figure 6D – According to this figure the authors showed that the amount of Fol in Fol-infected plants was only three folds of the mock infected plants (that are not supposed to contain any Fol at all!!!) which seems unreasonable. How specific are the IGS primers? Is it possible that they are also amplifying other fungal endophytes in the plant? Can the authors repeat this experiments with a different set of primers?

-Figure 5 – the presence of the sdRNA in the stem and the root should be monitored by RT-PCR or qRT-PCR as well.

Reviewer #2: • In the paper, the authors claim that the application of FolRDR1-dsRNAs on the surface of pre-infected tomato seedling leaves significantly inhibited the development of Fusarium wilt. Since FolRDR1-RNAi-SIGS works like a bio-pesticide, what are results of applying FolRDR1-dsRNAs before Fol infection? Whether it can provide better protection in the lab?

Reviewer #3: 1. The knockout experiments indicated that the FolRDR1 is required for the vegetative growth and asexual reproduction in Fol (Figure 1-2). As the authors mentioned that RDRs interact with RNAi machinery and sense transgene-induced silencing to generate dsRNA molecules in fungi (Line 95-103, 266-269), whether the function of RNA silencing pathway and accumulation of sRNA was affected in the FolRDR1-KO strains?

Koch et al. (2016) demonstrated that the F. graminearum DCL1 protein is required for exogenous dsRNA processing and intact fungal silencing machinery was required for SIGS in F. graminearum. If the FolRDR1 is essential for sRNA accumulation, when FolRDR1 was silenced by exogenous FolRDR1-dsRNA, this would in turn interfere with the own RNAi process of Fol. In order to ensure the efficiency of SIGS, the function of dsRNA targeted genes in the pathogen RNAi silencing process should be carefully considered. I suggest that small RNA sequencing analysis or northern blot should be carried out to explore the function of FolRDR1 and confirm the intact RNAi machinery of the FolRDR1-KO strains.

2. Line 120-125. The authors found that FolRDR1 was highly conservative in different F. oxysporum races, it is necessary to understand its conservation across species for a functional important protein. And the authors should make clear that how many RDR genes are present in the genome of Fol, and why FolRDR1 was chose for knockout.

3. The authors proved that artificial FolRDR1-dsRNAs can be taken up by Fol, and the mRNA level of RDR1 is repressed (Fig3). I suggest using Northern blot to detect the corresponding RDR1-sRNAs accumulated in the fungal mycelia to demonstrate that the dsRNA were processed into sRNAs in Fol.

4. The fluoresce signals observed in the undaubed leaves were weak and uneven in fig5C, the brighter spots look like autofluorescence from damaged leave cells. The quality of the pictures should be improved. In the stem vascular tissue (Fig5D), whether fluorescence signals in phloem and xylem can be distinguished with higher magnification. In addition, why the fluoresce signals were rich in the root hair in fig5E? And most importantly, whether the FolRDR1-sRNAs accumulated in tomato roots after taking up the FolRDR1-dsRNAs from leaves?

5. Spray of FolRDR1-dsRNAs alleviated the development of tomato wilt disease in fig6. However, the authors just observed the development of tomato wilt disease, did not monitor the accumulation of the FolRDR1-sRNAs in the infected tomato, neither detect the transcript level of FolRDR1 gene during infection. And I think it’s important to verify that the FolRDR1-dsRNAs or the FolRDR1-sRNAs were taken up by the Fol, and processed in the mycelia to induce the RNAi in the pathogen. Only then can it be termed cross-kingdom RNAi.

6. Many results lack experimental support in the Materials and methods section and the experimental conditions are not well described. Some examples, (1) Line 170, 179, 210, 230, 376-377, and 380-382, the concentration or the amount of the dsRNAs used? (2) Line 198-202, what’s the experimental conditions set to explore the optimal treatment in fig4D and fig4E. (3) How long the Fol took to produce conidia? And how to assess the production of the conidia in fig1D and fig4C? (4) Line 229-231, How long a gap should take to spray the dsRNA after inoculation of Fol? (5) For all the inoculation experiments, how many tomato seedlings were inoculated in one experiment, there was no statistics on the severity of the disease in fig 2C and fig6.

**Part III – Minor Issues: Editorial and Data Presentation Modifications**

Reviewer #1: -lines 227-255/figure 6 – how long after dsRNA treatment were the plants infected with Fol?

-How long after spraying with dsRNA were the leaves collected for RNA-seq analysis? dsRNA was previously suspected to act as an immune elicitor. The fact that neither dsGFP nor dsRDR1 caused any significant transcriptional changes in the plant is a bit surprising, unless the samples were collected long after spray application.

-Please provide a table of all the DEGs which were identified in the RNAseq analyses and their fold change.

-All figure legends – the number of times the experiments were repeated and the number of biological replicates in each experiment should be stated in the figure legends.

-Fig 2C – please split this section into three: Fig 2C - tomato pictures, Fig 2D – stem staining, Fig 2E –outgrowth assay.

-Figure S1B – why tomato actin was used for the tree construction? Is it a mistake?

-Figure S2B – The top (WT) and bottom (mutants) bands in the left gel are too close together and it's hard to really distinguish the size difference. Please repeat the experiment the run the gel longer.

-Table S2 - Please change the table title from "Table S1" to "Table S2"

-Line 145 – sentence is not clear. Please rephrase it.

-Line 212- remove the word "at"

-Line 235-236 – please rephrase the sentence

-Line 285-286 - please rephrase the sentence

Reviewer #2: 1) Please label the statistical analysis results in Figures 4C, D, E, and 6D.

2) In Figure 6A, the 15 d next to the bottom panel should be 25 d.

3) Line 238-241, please analyze the interpret the data in Figure 6B-D.

4) Please add the catalog number for the agents used in work.

5) Line 305 should be seedling leaves.

Reviewer #3: 1. Line 231-238, in the text the symptoms were described at 7 dpi and 25 dpi, while it was 15d in the figure 6A, and 2 weeks in the figure legend (Line 588). The descriptions must be consistent.

2. FolRDR1-dsRNA1 and FolRDR1-dsRNA2 was designed as the first half and second half of the RDR1 gene respectively according to Fig S5, I suggest that the location of the qRT-PCR primers should be labeled on the sequence to demonstrate the reliability of the transcription level of RDR1 detected by qRT-PCR.

PLOS authors have the option to publish the peer review history of their article (what does this mean?). If published, this will include your full peer review and any attached files.

Reviewer #1: No

Reviewer #2: No

Reviewer #3: **Yes: **Guiyan Huang
---

## [Decision Letter · Decision Letter 1]

24 Apr 2023

Dear Prof. Ouyang,

Thank you very much for submitting your manuscript "Artificial trans‐kingdom RNAi of FolRDR1 is a potential strategy to control tomato wilt disease" for consideration at PLOS Pathogens. As with all papers reviewed by the journal, your manuscript was reviewed by members of the editorial board and by several independent reviewers. The reviewers appreciated the attention to an important topic. Based on the reviews, we are likely to accept this manuscript for publication, providing that you modify the manuscript according to the review recommendations.

The reviewers commended the thorough effort in addressing the questions of 3 reviewers. Please address the request by Reviewer 3 on the trans‐kingdom RNAi in Fol. In addition, both reviewers 2 and 3 had some other minor issues that need to be addressed.

Sincerely,

Nian Wang

Academic Editor

PLOS Pathogens

Shou-Wei Ding

Section Editor

PLOS Pathogens

Kasturi Haldar

Editor-in-Chief

PLOS Pathogens

orcid.org/0000-0001-5065-158X

Michael Malim

Editor-in-Chief

PLOS Pathogens

orcid.org/0000-0002-7699-2064

The reviewers commended the thorough effort in addressing the questions of 3 reviewers. Please address the request by Reviewer 3 on the trans‐kingdom RNAi in Fol. In addition, both reviewers 2 and 3 had some other minor issues that need to be addressed.

Reviewer Comments (if any, and for reference):

Reviewer's Responses to Questions

**Part I - Summary**

Reviewer #1: The authors properly addressed all issues that were raised in the first review cycle.

The following small cosmetic adjustments should by added in the proof prior to publication:

*All figure legends - Please change "biological replicates in each…" to "biological replicates were used in each…"

*Figures 6 and S8– the sentence "presents the largest average, de present significance…" is unclear. Please rephrase.

*Figure S2C - "C" marking is missing. 1 kb and 2 kb markings in the DNA ladder were accidentally switched. Please fix it.

*Line 181 – please change "(Fig 2C)" to "(fig 2C-E)"

*Lines 240-251 – As requested, the authors also checked the presence of dsRNA in the stem and root by RT-PCR in addition to the labeled RNA (Fig 5A). This should be addressed in the text as well.

Reviewer #2: The author mentioned that they performed both pre-treatment and after-treatment with FolRDR1-dsRNAs experiments and found that no significant difference was shown between these two treatments. Please show the data of pre-treatment with FolRDR1-dsRNAs experiment in the supplementary figures.

Reviewer #3: (No Response)

**Part II – Major Issues: Key Experiments Required for Acceptance**

Reviewer #1: (No Response)

Reviewer #2: (No Response)

Reviewer #3: To demonstrate the trans‐kingdom RNAi in Fol, it is necessary not only to prove the RDR1-dsRNAs were taken up by Fol, but also that the dsRNAs were processed into small interfering (si)RNAs. The authors proved that artificial FolRDR1-dsRNAs can be taken up by Fol (Fig3) and tomato (Fig5) using confocal microscopy laser scanner and RT-PCR. But there was no evidence of the accumulation of corresponding RDR1-dsRNA-derived siRNAs. Fig 3E and Fig 5A were updated to show the corresponding mRNA levels of FolRDR1 in the fungal mycelia and FolRDR1-dsRNAs transferred from treated leaves to stems and roots. However, these results did not prove that FolRDR1-dsRNAs were processed into siRNAs to induce RNAi in Fol. The authors may have misunderstood my questions. I suggested using Northern blot to detect the RDR1-dsRNA-derived siRNAs accumulated in the fungal mycelia (Fig3) and in the infected tomato (Fig6).

**Part III – Minor Issues: Editorial and Data Presentation Modifications**

Reviewer #1: (No Response)

Reviewer #2: (No Response)

Reviewer #3: The authors replied that for all the inoculation experiments, at least 30 tomato seedlings were inoculated for each treatment. However, it was “three biological replicates in each experiment” in the revised figure legends of fig 2 and fig 6. Please be consistent in your description of the experimental data.

PLOS authors have the option to publish the peer review history of their article (what does this mean?). If published, this will include your full peer review and any attached files.

Reviewer #1: No

Reviewer #2: No

Reviewer #3: No

Figure Files:

Data Requirements:

Reproducibility:

References:

---

## [Decision Letter · Decision Letter 2]

5 Jun 2023

Dear Prof. Ouyang,

We are pleased to inform you that your manuscript 'Artificial trans‐kingdom RNAi of FolRDR1 is a potential strategy to control tomato wilt disease' has been provisionally accepted for publication in PLOS Pathogens.

Best regards,

Nian Wang

Academic Editor

PLOS Pathogens

Shou-Wei Ding

Section Editor

PLOS Pathogens

Kasturi Haldar

Editor-in-Chief

PLOS Pathogens

orcid.org/0000-0001-5065-158X

Michael Malim

Editor-in-Chief

PLOS Pathogens

orcid.org/0000-0002-7699-2064

We thank the authors for the diligent effort in addressing the concerns of the reviewers and congratulate the authors for the excellent work.

Reviewer Comments (if any, and for reference):

Reviewer's Responses to Questions

**Part I - Summary**

Reviewer #3: (No Response)

**Part II – Major Issues: Key Experiments Required for Acceptance**

Reviewer #3: (No Response)

**Part III – Minor Issues: Editorial and Data Presentation Modifications**

Reviewer #3: (No Response)

PLOS authors have the option to publish the peer review history of their article (what does this mean?). If published, this will include your full peer review and any attached files.

Reviewer #3: No

---

## [Editor Report · Acceptance letter]

13 Jun 2023

Dear Prof. Ouyang,

We are delighted to inform you that your manuscript, "Artificial trans‐kingdom RNAi of FolRDR1 is a potential strategy to control tomato wilt disease," has been formally accepted for publication in PLOS Pathogens.

Best regards,

Kasturi Haldar

Editor-in-Chief

PLOS Pathogens

orcid.org/0000-0001-5065-158X

Michael Malim

Editor-in-Chief

PLOS Pathogens

orcid.org/0000-0002-7699-2064